# Morphometric Measurement of Mean Cortical Curvature: Analysis of Alterations in Cognitive Impairment

**DOI:** 10.3390/medicina61030531

**Published:** 2025-03-18

**Authors:** Renāte Rūta Apse, Nauris Zdanovskis, Kristīne Šneidere, Guntis Karelis, Ardis Platkājis, Ainārs Stepens

**Affiliations:** 1Department of Radiology, Riga Stradins University, LV-1007 Riga, Latvia; renateruta.apse@rsu.lv (R.R.A.); ardis.platkajis@rsu.lv (A.P.); 2Department of Radiology, Riga East University Hospital, LV-1038 Riga, Latvia; 3Institute of Public Health, Riga Stradins University, LV-1007 Riga, Latvia; kristine.sneidere@rsu.lv (K.Š.); ainars.stepens@rsu.lv (A.S.); 4Department of Health Psychology and Pedagogy, Riga Stradins University, LV-1007 Riga, Latvia; 5Department of Neurology and Neurosurgery, Radiology, Riga East University Hospital, LV-1038 Riga, Latvia; guntis.karelis@rsu.lv; 6Department of Infectiology, Riga Stradins University, LV-1007 Riga, Latvia

**Keywords:** atlas-based segmentation, cognitive impairment, dementia, mean cortical curvature, MoCA score, neuroimaging, structural MRI

## Abstract

*Background and Objectives*: Cognitive impairment, including mild cognitive impairment (MCI) and Alzheimer’s disease (AD), is a growing public health concern. Early detection and an understanding of structural changes are crucial for accurate diagnosis and timely intervention. Cortical curvature, a morphometric measure derived from structural magnetic resonance imaging (MRI), has emerged as a potential biomarker for neurodegenerative processes. This study investigates the relationship between mean cortical curvature and cognitive impairment. *Materials and Methods*: A cross-sectional study was conducted with 58 participants, categorized into, first, cognitively impaired (CI) and non-cognitively impaired (NC) groups and, second, a normal cognitive group (NC), a mild cognitive performance group (MPG), and a low cognitive performance group (LPG) based on the Montreal Cognitive Assessment (MoCA) score. MRI data were acquired using a 3.0 Tesla scanner, and cortical reconstruction was performed using FreeSurfer 7.2.0. Mean cortical curvature values were extracted for 34 brain regions per hemisphere. *Results*: Significant differences in mean cortical curvature were found between the CI and NC groups. In the right hemisphere, statistically significant changes in mean curvature were observed in the isthmus cingulate (U = 188.5, *p* = 0.006), lingual (U = 202.5, *p* = 0.013), pars orbitalis (U = 221.5, *p* = 0.031), and posterior cingulate regions (U = 224.5, *p* = 0.035). In the left hemisphere, significant differences were detected in the cuneus (U = 226.5, *p* = 0.038) and posterior cingulate (U = 231.5, *p* = 0.046) regions. Analysis across three cognitive performance groups (NC, MPG, and LPG) showed significant curvature differences in the right isthmus cingulate (H(2) = 7.492, *p* = 0.024) and lingual regions (H(2) = 6.250, *p* = 0.044). *Conclusions*: Decreased mean cortical curvature in brain regions associated with cognitive function could be indicative of cognitive impairment and may reflect early neurodegenerative changes. These results highlight cortical curvature as a potential structural sign for cognitive impairment, showing the need for further investigation in longitudinal studies.

## 1. Introduction

Cognitive impairment, encompassing conditions such as mild cognitive impairment or neurocognitive disorders such as Alzheimer’s disease (AD), poses significant challenges to public health due to its increasing prevalence as the population ages and its impact on individuals’ quality of life [1,2]. Early detection and understanding of the underlying neural mechanisms are crucial for developing effective interventions if possible and further assistance to the patients and their relatives [3]. Neuroimaging studies have utilized structural magnetic resonance imaging (MRI) to investigate brain morphometry, including cortical thickness, curvature measurements, surface area, brain volume, and more, in cognitive disorders [4,5,6]. Among various morphometric measures, cortical curvature has emerged as a potential biomarker for neurodegenerative diseases [7].

While techniques such as structural MRI for volumetric analysis, functional MRI (fMRI) for functional connectivity, diffusion tensor imaging (DTI) for white matter integrity, and proton magnetic resonance spectroscopy for metabolic analysis provide valuable insights into cognitive disorders, there is an increasing potential in morphometric measures, one of which is cortex curvature [8,9,10].

Cortical curvature refers to the degree of folding of the cerebral cortex, and it is further explained in Figure 1.

Studies have indicated that the cortical curvature may show early cytoarchitectural changes [12,13,14], which could serve as an early biomarker for cognitive impairment. Changes in cortical folding patterns have been associated with aging and neurodegenerative processes. For instance, a study by De Moraes et al. (2024) [7] demonstrated that changes in cortical folding correlate with the progression of Alzheimer’s disease, suggesting that cortical curvature could serve as an indicator of disease progression.

The cortical curvature as a measurement is interrelated with brain volume and atrophy. One study analyzed patients with mild traumatic brain injury and showed that 50% bilateral cortical regions had an increased mean cortical curvature [15]. These alterations in curvature related to reductions in both gray and white matter volume, suggesting that regional atrophy contributed to cortical surface deformations [15]. Similarly, studies on healthy aging have found that while cortical thickness and white matter volume decrease with aging, surface area and mean curvature remain relatively stable, further showing that atrophy affects cortical thickness rather than the surface morphology [16]. Research on brain size and cortical structure further reveals that as brain size increases, the cortex thickens but very minimally, while the sulcal convolution increases significantly [17]. These findings highlight the complex and still not fully understood relationship between brain atrophy, volume changes, and alterations in mean cortical curvature.

By using advanced software tool such as FreeSurfer, cortical curvature analysis enables a detailed examination of curvature patterns across different brain regions. While previous neuroimaging techniques have primarily focused on volumetric and functional connectivity, cortical curvature may provide complementary information for early detection and potentially tracking of cognitive decline [7]. A better understanding of these morphometric changes may help in complementing diagnostic criteria, enabling earlier detection of neurodegenerative disorders. As a result, this could facilitate more timely and targeted interventions, such as cognitive therapies or pharmacological treatments, as a result improving patient outcomes.

Regional variations of cortical curvature may be linked to specific cognitive functions. Research has shown that certain brain regions such as the cingulate region, the entorhinal cortex, and the orbital frontal cortex exhibit distinct curvature patterns that correlate with cognitive performance [8,18]. For instance, differences in mean curvature have been observed in regions implicated in memory and executive functions, which are often affected in mild cognitive impairments [7,19].

Despite the studies that have already touched upon curvature analysis of the brain and its potential relation to cognitive impairments, this relationship between both variables remains complex. There are studies that have found significant associations between decreased mean curvature and cognitive decline; however, there are not many studies that have investigated this field, and furthermore, it is important to take into consideration methodology differences in the existing studies [7,14]. For instance, differences in methodology—such as in the study by McCarthy (2015), which compared FreeSurfer-generated output with and without manual intervention [20]—revealed that manual corrections led to differences, particularly in brain regions primarily affected by Alzheimer’s disease and schizophrenia.

In this context, this study aims to investigate whether mean cortical curvature can potentially serve as a biomarker for distinguishing cognitively impaired individuals from healthy controls. Specifically, we sought to determine if alterations in cortical curvature are associated with different levels of cognitive performance and whether these changes correlate with established measures of cognitive decline. Understanding these structural brain changes is essential for developing targeted interventions and improving diagnostic accuracy in cognitive disorders.

## 2. Materials and Methods

This study was conducted as a cross-sectional study, and it included 58 participants. Initially, participants were divided into two groups—those who were cognitively impaired (CI) (MoCA score ≤ 25) and those who did not have cognitive impairment, i.e., normal cognition (NC) (MoCA score ≥ 26)—to broadly assess differences in cortical curvature between individuals with and without cognitive impairment. However, given that cognitive decline occurs on a spectrum, a second classification was applied, dividing participants into three groups: normal cognition (NC), moderate cognitive performance (MPG), and low cognitive performance (LPG) based on MoCA scores. This finer categorization allowed for a more detailed analysis of cortical curvature changes across varying degrees of cognitive impairment, providing insights into potential alterations in brain morphology. An overview of participant age, sex, and MoCA scores is visualized in Table 1.

A Chi-square test was performed on sex between the groups, and the results showed that there was a statistically significant difference between both groups (χ^2^ = 5.007, *p* = 0.025). To analyze differences in age and MoCA scores between the two groups, the Mann–Whitney U test was performed. The results indicated that there was no statistically significant difference in age between the groups (U = 367.500, *p* = 0.752). However, a statistically significant difference was found in MoCA scores (U = 0.000, *p* < 0.001), confirming distinct cognitive performance levels between the groups.

For further analysis the patients were divided into three groups: the normal cognition group (NC) with MoCA scores ≥26, the moderate cognitive performance group (MPG) with MoCA scores ≥15 and ≤25, and the low cognitive performance group (LPG) with MoCA scores ≤14 [21]. An overview of participant age, sex, and MoCA scores is visualized in Table 2.

A Chi-square test on sex was performed and did not show statistical significance among the three groups (X^2^ = 5.013, *p* = 0.082). To determine whether there were any statistically significant differences for age and MoCA scores among three groups, a Kruskal–Wallis test was performed and showed that there were statistically significant differences among groups for MoCA scores (H(2) = 48.176, *p* < 0.001); by performing Dunn’s post hoc test, there were statistically significant differences found between MPG-NC groups (*p* < 0.001 after Bonferroni and Holm correction statistical significance was maintained), MPG-LPG groups (*p* < 0.001 after Bonferroni and Holm correction statistical significance was maintained) and NC-LPG (*p* < 0.001 after Bonferroni and Holm correction statistical significance was maintained). With regards to age, there was no statistical significance among the three groups (H(2) = 0.111, *p* = 0.964).

### 2.1. Participant Selection

As part of this study, individuals were referred for assessment by a board-certified neurologist with an expertise in cognitive impairment due to either self-reported cognitive concerns or suspected cognitive decline. All participants selected for inclusion were right-hand dominant.

The exclusion criteria included significant neurological disorders such as brain tumors, major strokes, intracerebral lobar hemorrhages, congenital malformations, Parkinson’s disease, and multiple sclerosis. Additionally, individuals with a history of substance or alcohol abuse, major depressive disorder, schizophrenia, or other severe psychiatric conditions were excluded. Those with documented significant vascular diseases were also not eligible for participation. Patients were included if MRI findings did not indicate any other substantial pathological abnormalities among the study participants.

### 2.2. MRI Data Acquisition and Analysis

MRI was performed using a 3.0 Tesla MRI scanner (GE Healthcare, Mumbai, MA, USA) in a university hospital setting. For post-processing, 3D T1-weighted axial images were acquired using the following parameters: flip angle of 11, TE set to minimum full, TI of 400, field of view (FOV) of 25.6, and a slice thickness of 1 mm. Additional imaging sequences, including T2, 3D FLAIR, DWI, ADC, and SWI, were utilized to exclude other clinically significant pathologies.

### 2.3. Data Parcellation and Segmentation

Cortical reconstruction was carried out using FreeSurfer 7.2.0 image analysis software, which is well documented and freely available for download at http://surfer.nmr.mgh.harvard.edu/, accessed on 7 February 2025. Further descriptions on morphometric analysis used within the FreeSurfer program are published on the FreeSurfer website. The methodological specifics of these procedures have been detailed in previous publications [11,15,22,23,24,25,26]. For data parcellation, FreeSurfer uses two main atlases—the Desikan–Killiany cortical atlas, which is gyral based, and the Destrieux atlas, which looks more into the gyri and banks of sulci and ultimately provides information on curvature [27,28].

### 2.4. Statistical Analysis

Statistical analyses were conducted using JASP 0.19.1 (Eric-Jan Wagenmakers, Amsterdam, The Netherlands) [29]. The analysis included descriptive statistics, the Chi-square test, the Mann–Whitney U test, the Kruskal–Wallis test, and Dunn’s post hoc analysis. These tests were chosen due to the non-parametric nature of the data—the initial assessment of data normality was performed using the Shapiro–Wilk test for age and MoCA score, which indicated that non-parametric tests were most appropriate to use; furthermore, each group consisted of *n* < 30, which also was an indication to use a non-parametric test.

Descriptive statistics were used to summarize general variables and assess differences between groups. The Chi-square test was applied to examine associations between categorical variables. The Mann–Whitney U test was used to determine statistically significant differences between individuals without cognitive impairment and those with impaired cognitive function, as assessed by the MoCA. The Kruskal–Wallis test was utilized to assess significant differences among the NC, MPG, and LPG groups. When significant differences were detected, Dunn’s post hoc test was conducted to determine pairwise group differences, with additional Bonferroni and Holm corrections applied to mitigate the risk of type I errors (false positive).

## 3. Results

### 3.1. Statistical Differences Between Two Groups—NC and CI

Overall, 34 brain regions and their mean curvatures were analyzed for both the right and the left hemisphere. When looking at two groups—NC and CI—the Mann–Whitney U test was performed. Statistically significant changes were found in the right hemisphere isthmus cingulate region (*p* = 0.006), the lingual region (*p* = 0.013), the pars orbitalis region (*p* = 0.031), and the posterior cingulate region (*p* = 0.035), while within the left hemisphere, statistically significant changes among the two groups were found in the cuneus region (*p* = 0.038) and the posterior cingulate region (*p* = 0.046). However, this part served as an exploratory test as there were no multiple comparison corrections applied. The results are visualized in Table 3.

The mean curvature values of the regions that showed statistical difference are further presented in Table 4.

### 3.2. Statistical Differences Between Three Groups—NC, MPG, and LPG

To analyze the statistical differences between three groups, the Kruskal–Wallis test was performed.

Statistical differences were found in the following:
Right hemisphere isthmus cingulate region—H(2) = 7.492, *p* = 0.024 (see Figure 2). The anatomical location is represented in Figure 3.

Further testing with Dunn’s post hoc test showed statistically significant differences between the MPG and NC groups (*p* = 0.009). After Bonferroni and Holm correction, the statistical significance remained unchanged (Table 5).

2.Right hemisphere lingual region—H(2) = 6.250, *p* = 0.044 (see Figure 4). The anatomical location is represented in Figure 5.

Dunn’s post hoc test showed statistically significant difference between groups MPG and NC and LPG and NC; however, the statistical differences did not persist after Bonferroni and Holm corrections (Table 6).

3.Right hemisphere pars orbitalis region—H(2) = 6.261, *p* = 0.044 (see Figure 6). The anatomical location is represented in Figure 7.

Dunn’s post hoc test showed a statistically significant difference between groups MPG and NC (*p* = 0.014, and after Bonferroni and Holm correction the statistical significance remained unchanged) (Table 7).

There were no statistically significant differences among the three groups in the region of posterior cingulate in the right hemisphere and the regions located in the left hemisphere—the cuneus and posterior cingulate.

## 4. Discussion

While brain morphometric biomarkers are still at a comparatively early stage of research, they could indicate cognitive impairment at early stages to employ interventions where possible. In recent years, cortical curvature has gained more attention as a potential neuroimaging marker that extends beyond the traditional measures such as cortical thickness and volume analysis [30,31]. This study further looked at the association between cortical curvature and cognitive impairment (based on MoCA score results), identifying the brain regions where mean curvature reductions were statistically significantly linked to cognitive impairment. This section examines this study’s findings in relation to previous research and outlines potential future research directions.

This study provides analysis of specific cortical region mean curvature alterations and their association with cognitive impairment, particularly in regions involved in memory consolidation, executive functions, and visuospatial processing. The results demonstrated that there were statistically significant differences among cognitively impaired patients in the posterior cingulate, isthmus cingulate, lingual, pars orbitalis, and cuneus regions. These findings aligned with previous research showing that changes in cortical folding may serve as an early neurodegenerative indicator [7]. Specifically, the posterior cingulate region plays a crucial role in memory retrieval and attention [32].

Similarly, structural changes in the isthmus cingulate and lingual regions suggested disruptions in memory-related and visual spatial processing function [7,33]. The isthmus cingulate region is linked to the hippocampus and medial temporal lobe and is susceptible to neurodegeneration [34]. On the other hand, the lingual region, associated with visual memory and object recognition, has been indicated in an early-stage cognitive decline [35]. Thus, the structural changes could potentially contribute to functional impairments manifesting as early cognitive decline.

Additionally, the pars orbitalis region that is part of the inferior frontal gyrus also presented curvature changes with cognitive impairment. This region plays a role in executive functioning, verbal fluency, and decision making [36]. The findings of this study were consistent with prior research showing that frontal lobe involvement in cognitive impairment correlated with deficits in working memory and inhibitory control [37].

Several mechanisms that could explain the changes in cortical curvature were observed in this study. One hypothesis, known as the tension-based folding theory, demonstrates that cortical gyrification is driven by axonal tension with underlying white matter tracts [38]. Disruption in the integrity of white matter, as is often observed in cognitive decline, can lead to a loss of cortical tension and thus reduction in gyrification and flattening of the cortex, which subsequently alters the curvature of the cortex [39].

Another theory that could explain the changes is the synaptic density, which assumes that cortical curvature is related to the density and organization of neuronal synapses [40]. High-curvature regions are more likely to exhibit significant synaptic density, and neurodegenerative mechanisms with lessening synaptic density and dentritic retraction could contribute to reductions in curvature over time [40]. This explanation is also in line with the evidence that high-curvature cortical areas are metabolically more active and thus more vulnerable to degeneration (e.g., amyloid-beta accumulation) [41]. For example, the posterior cingulate has one of the highest baseline metabolic rates in the brain, which makes it particularly sensitive to neurodegeneration [32,42].

This study’s results highlight the potential utility of mean cortical curvature as an additional tool for earlier detection of cognitive impairment. Traditional neuroimaging biomarkers, such as the hippocampal atrophy and ventricular enlargement, are often detected after significant neurodegenerative damage has occurred [43]. Mean curvature may detect subtle morphometric changes at an earlier stage. From a clinical perspective, incorporating mean curvature analysis into neuroimaging diagnostics may enhance accuracy in diagnosing MCI and early-stage Alzheimer disease. Furthermore, adding other biomarkers such as PET imaging for amyloid detection, cortical thickness, volumetric measurements, and functional connectivity measures could increase the sensitivity and specificity of diagnosing cognitive impairment [44,45].

While this study provides insights into potential additional biomarkers for cognitive impairment, there are several limitations that have to be taken into consideration. First, the cross-sectional study design does not allow further conclusions regarding the dynamics of cortical curvature changes. A longitudinal study would be useful in determining the prediction or risk of future cognitive decline and progression of disease. Second, the sample size is relatively small, which further limits generalizing our findings. A study with a larger and more diverse population could confirm and potentially provide further insights into the viability of mean curvature as a biomarker of cognitive decline, especially for early cognitive decline. Third, this study focused on structural alterations; however, performing also DTI and fMRI could give more insights in understanding the relationship between cortical mean curvature and cognitive decline.

## 5. Conclusions

Our study provided evidence that reductions in mean cortical curvature in the posterior cingulate, isthmus cingulate, and lingual regions may be associated with cognitive impairment. These findings reinforced the importance of cortical folding patterns in neurodegenerative disease progression. However, while cortical curvature metrics showed significant differences at the group level, their variability limited their use as standalone biomarkers for individual diagnoses. Future research should investigate whether combining cortical curvature with other neuroimaging or cognitive biomarkers can enhance early detection at the individual level.

While there are limitations to this study, it shows the potential and the need to further investigate morphometric measurement such as mean curvature to detect cognitive decline early on. As indicated beforehand, there should be a larger cohort of participants in the longitudinal study to further ground the results that have been shown as part of this study.

## Figures and Tables

**Figure 1 medicina-61-00531-f001:**
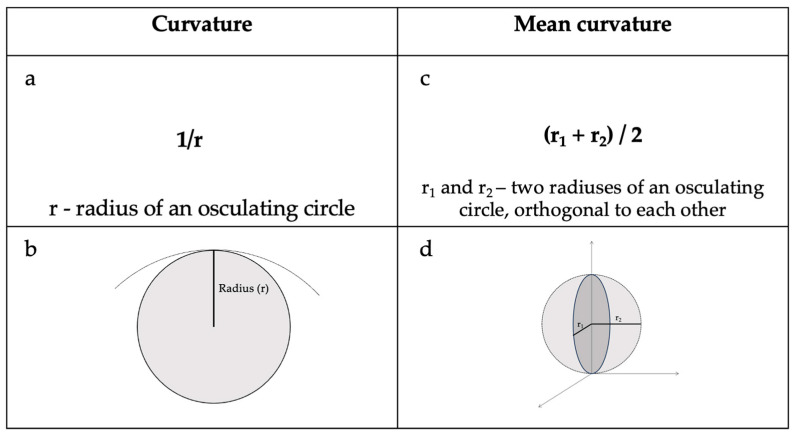
(**a**) Curvature equation and (**b**) radius of an osculating circle at a point. (**c**) Mean curvature equation and (**d**) principal radius and their orthogonal location in relation to each other [11].

**Figure 2 medicina-61-00531-f002:**
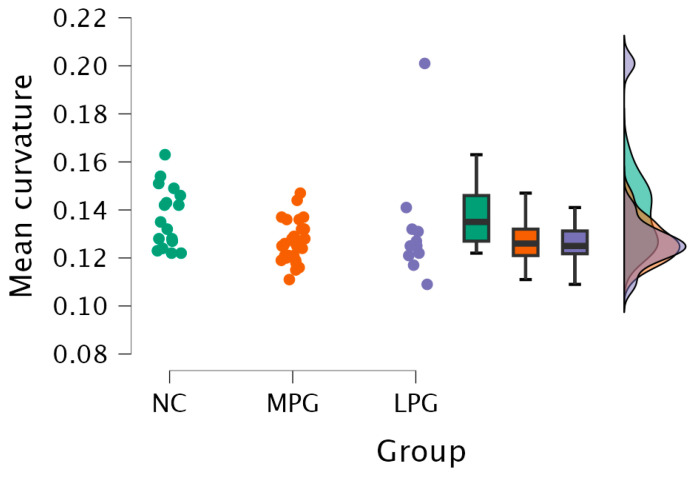
Mean curvature differences in the isthmus cingulate region in the right hemisphere in all three groups: NC, MPG, and LPG.

**Figure 3 medicina-61-00531-f003:**
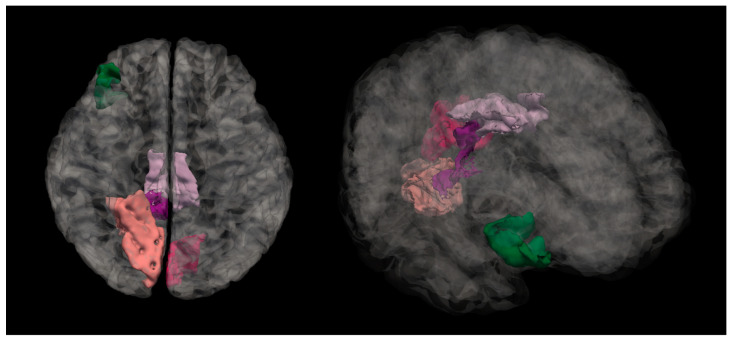
Anatomical location of the isthmus cingulate region. The highlighted dark purple region represents the area where statistically significant differences in the mean cortical curvature were observed between groups.

**Figure 4 medicina-61-00531-f004:**
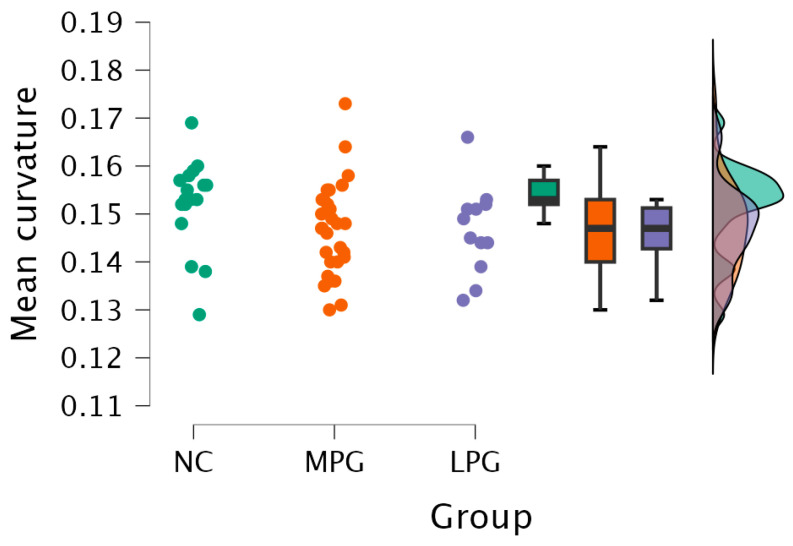
Mean curvature differences in the lingual region in the right hemisphere in all three groups: NC, MPG, and LPG.

**Figure 5 medicina-61-00531-f005:**
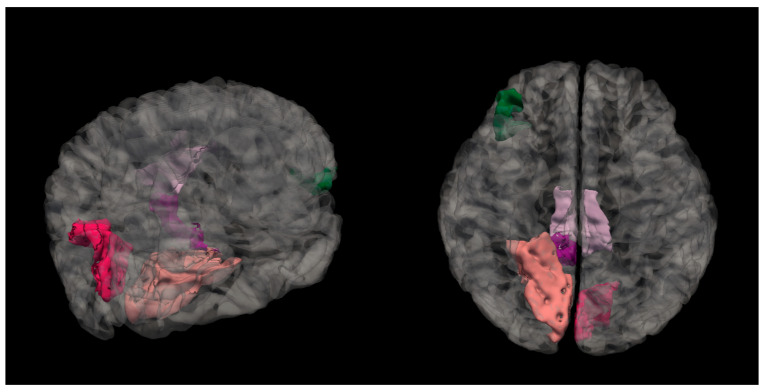
Anatomical location of the lingual region in the right hemisphere, highlighted in salmon pink. This region showed significant differences in mean cortical curvature across the cognitive performance groups (NC, MPG, and LPG).

**Figure 6 medicina-61-00531-f006:**
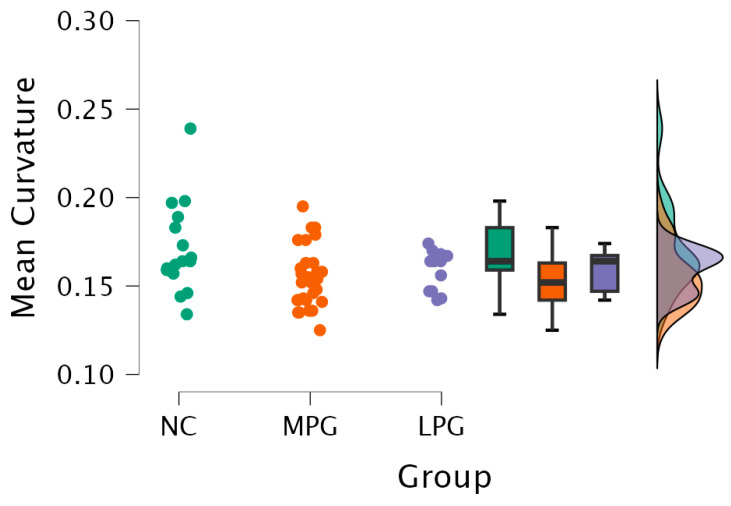
Mean curvature differences in pars orbitalis region in the right hemisphere in all three groups: NC, MPG, and LPG.

**Figure 7 medicina-61-00531-f007:**
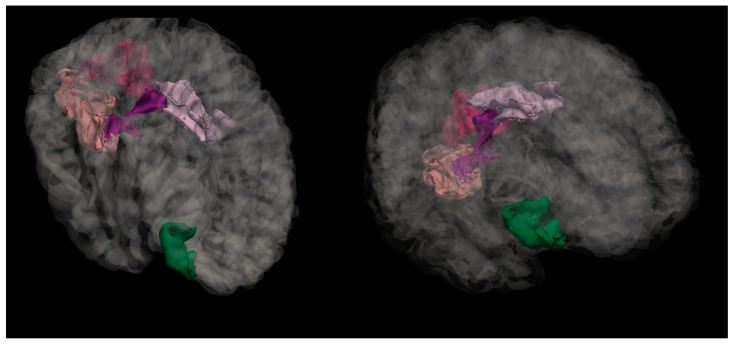
Anatomical location of the pars orbitalis region, shown in green. This area, part of the inferior frontal gyrus, exhibited significant differences in mean cortical curvature between groups.

**Table 1 medicina-61-00531-t001:** Age, sex distribution, and MoCA scores among participants in this study (divided into two groups).

	Age		MoCA		Sex (F:M)	
	NC	CI	NC	CI	NC	CI
Valid	17	41	17	41	17	41
Mean	70.941	73.098	27.529	18.146	16:1	27:14
Standard deviation	7.554	8.297	1.231	6.916		
Minimum	51.000	57.000	26.000	4.000		
Maximum	83.000	96.000	30.000	25.000		
χ^2^					5.007 (*p* = 0.025)	
Mann–Whitney U test	367.500 (*p* = 0.752)		0.000 (*p* < 0.001)			

NC—normal cognition; CI—cognitive impairment.

**Table 2 medicina-61-00531-t002:** Age, sex distribution, and MoCA scores among participants in this study (divided into three groups).

		Age			MoCA		Sex (F:M)		
	NC	MPG	LPG	NC	MPG	LPG	NC	MPG	LPG
Valid	17	29	12	17	29	12	16:1	19:10	8:4
Mean	70.942	72.345	74.917	27.529	22.103	8.583			
Standard deviation	7.554	7.262	10.535	1.231	2.650	3.872			
Minimum	51.000	57.000	62.000	26.000	15.000	4.000			
Maximum	83.000	85.000	96.000	30.000	25.000	14.000			
χ^2^							5.013 (*p* = 0.082)		
H(2)	0.111 (*p* = 0.946)			48.176 (*p* < 0.001)					

NC—normal cognitive performance group; MPG—moderate cognitive performance group; LPG—low cognitive performance group.

**Table 3 medicina-61-00531-t003:** Mann–Whitney U test for each brain region in the right and left hemispheres among two groups.

Brain Region	U	*p*
RH banks of the superior temporal	282	0.259
RH caudal anterior cingulate	247.5	0.086
RH caudal middle frontal cingulate	415	0.259
RH cuneus	301	0.422
RH entorhinal	247.5	0.086
RH fusiform	341	0.905
RH interior parietal	263.5	0.149
RH inferior temporal	315	0.573
RH isthmus cingulate	188.5	0.006
RH lateral occipital	333	0.798
RH lateral orbitofrontal	307.5	0.489
RH lingual	202.5	0.013
RH medial orbitofrontal	259	0.128
RH middle temporal	240	0.065
RH parahippocampal	311	0.527
RH paracentral	308	0.494
RH pars opercularis	266.5	0.164
RH pars orbitalis	221.5	0.031
RH pars triangularis	300.5	0.417
RH pericalcarine	248.5	0.089
RH postcentral	346.5	0.98
RH posterior cingulate	224.5	0.035
RH precentral	315	0.573
RH precuneus	300.5	0.416
RH rostral anterior cingulate	272.5	0.197
RH rostral middle frontal	292	0.338
RH superior frontal	264.5	0.154
RH superior parietal	363	0.811
RH superior temporal	257	0.12
RH supramarginal	260.5	0.135
RH frontal pole	279	0.238
RH temporal pole	365	0.784
RH transverse temporal	303	0.524
RH insula	255.5	0.143
LH banks of the superior temporal	351.5	0.966
LH caudal anterior cingulate	252	0.101
LH caudal middle frontal cingulate	366	0.771
LH cuneus	226.5	0.038
LH entorhinal	404.5	0.343
LH fusiform	359	0.864
LH interior parietal	322.5	0.663
LH inferior temporal	283.5	0.27
LH isthmus cingulate	256	0.116
LH lateral occipital	319.5	0.626
LH lateral orbitofrontal	316.5	0.59
LH lingual	250	0.094
LH medial orbitofrontal	306	0.473
LH middle temporal	300.5	0.417
LH parahippocampal	347	0.986
LH paracentral	329.5	0.752
LH pars opercularis	364	0.798
LH pars orbitalis	331	0.771
LH pars triangularis	294	0.356
LH pericalcarine	276.5	0.222
LH postcentral	384.5	0.544
LH posterior cingulate	231.5	0.046
LH precentral	348	1
LH precuneus	316.5	0.59
LH rostral anterior cingulate	288.5	0.309
LH rostral middle frontal	283	0.267
LH superior frontal	278.5	0.234
LH superior parietal	372	0.694
LH superior temporal	334	0.811
LH supramarginal	333.5	0.804
LH frontal pole	341.5	0.912
LH temporal pole	276	0.219
LH transverse temporal	353	0.827
LH insula	251.5	0.125

RH—right hemisphere; LH—left hemisphere.

**Table 4 medicina-61-00531-t004:** Values of mean curvatures among the statistically significant regions.

	RH Isthmus Cingulate		RH Lingual		RH Pars Orbitalis		LH Cuneus		LH Posterior Cingulate	
	NC	CI	NC	CI	NC	CI	NC	CI	NC	CI
Valid	17	41	17	41	17	41	17	41	17	41
Mean	0.137	0.128	0.152	0.147	0.170	0.156	0.153	0.144	0.140	0.134
Std. deviation	0.013	0.014	0.009	0.009	0.025	0.016	0.015	0.013	0.011	0.013

**Table 5 medicina-61-00531-t005:** Dunn’s post hoc test of the three groups divided by MoCA scores and mean curvature in the right hemisphere isthmus cingulate region.

Comparison	z	W_i_	W_j_	r_rb_	*p*	P_bonf._	P_holm_
MPG–NC	−2.597	25.534	38.912	0.471	0.009	0.028	0.028
MPG–LPG	−0.037	25.534	25.750	0.006	0.970	1.000	0.970
NC–LPG	2.07	38.912	25.75	0.431	0.038	0.115	0.077

**Table 6 medicina-61-00531-t006:** Dunn’s post hoc test of the three groups divided by MoCA scores and mean curvature in the right hemisphere lingual region.

Comparison	z	W_i_	W_j_	r_rb_	*p*	P_bonf._	P_holm_
MPG−NC	−2.316	26.155	38.088	0.398	0.021	0.062	0.062
MPG−LPG	0.128	26.155	25.417	0.006	0.899	1.000	0.899
NC−LPG	1.992	38.088	25.417	0.471	0.046	0.139	0.093

**Table 7 medicina-61-00531-t007:** Dunn’s post hoc test of the three groups divided by MoCA scores and mean curvature in the right hemisphere pars orbitalis region.

Comparison	z	W_i_	W_j_	r_rb_	*p*	P_bonf._	P_holm_
MPG−NC	−2.459	24.293	36.971	0.438	0.014	0.042	0.042
MPG−LPG	−1.244	24.293	31.500	0.247	0.213	0.640	0.427
NC−LPG	0.860	36.971	31.500	0.186	0.390	1.000	0.427

## Data Availability

The raw data supporting the conclusions of this article will be made available by the authors on request.

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
