# Peer review of "Morphometric Measurement of Mean Cortical Curvature: Analysis of Alterations in Cognitive Impairment"

_medicina, 2025, doi:10.3390/medicina61030531_

Round 1
Reviewer 1 Report
Comments and Suggestions for Authors
Dear authors,
I have carefully reviewed your manuscript entitled "Morphometric Measurement of Mean Cortical Curvature: Analysis of Alterations in Cognitive Impairment." The subject matter is highly relevant, particularly given the increasing prevalence of cognitive disorders and their significant impact on both public health systems and affected individuals' families. Research in this field is crucial for improving early diagnosis, understanding disease progression, and ultimately developing more effective intervention strategies.
Your study protocol is well-conceived and meticulously structured, ensuring a solid scientific foundation. The methodology is described in detail, adhering closely to the established protocol and allowing for reproducibility of the study. The analytical approach is robust, and the selection of morphometric measurements, particularly mean cortical curvature, is a valuable contribution to the ongoing efforts to identify reliable neuroimaging biomarkers for cognitive impairment.
The results are presented clearly and concisely, effectively highlighting the observed alterations in cortical curvature associated with cognitive disorders. Additionally, the discussion section is well-developed, contextualizing the findings within the existing body of literature and offering insightful interpretations. The conclusions drawn are logically supported by the data and reinforce the study’s significance in advancing our understanding of structural brain changes in cognitive impairment.
Given the strong methodological framework, the clarity of the presentation, and the clinical relevance of the findings, I find the manuscript to be of high quality. I do not see the need for any major revisions, as the study is already well-structured and compelling in its current form.
Best regards.
Author Response
Dear Reviewer,
We sincerely appreciate your thoughtful and positive evaluation of our manuscript. Your recognition of the relevance of this research, as well as your assessment of our methodology, analytical approach, and presentation of findings, is encouraging.
Thank you for your time and consideration. We are pleased that our manuscript meets your expectations, and we look forward to its potential contribution to the scientific community.
Sincerely,
Renāte Rūta Apse
Reviewer 2 Report
Comments and Suggestions for Authors
The topic of the article is of interest to readers and can be published after revision. There are several comments and questions that need to be answered.
- Line 27: It is unclear which group (CI or NC) had a lower mean curvature.
- Line 67: There is no reference to this author and article in the References.
- Lines 87-89: It’s unclear how the understanding of morphometric alterations could potentially improve the intervention pathway. You should explain this phrase, or provide a
- Line 106: The aim should be specified. It should not be phrased as “further examine…”. It should be explained what were you trying to find exactly?
- Line 115: You divided the participants into two groups, but then you divided them into three groups. Could you explain why you did that?
- Line 116: What about patients with 25 scores?
- Lines 122-123: Does this phrase mean that there was no statistically significant difference in age?
- Table 1, Table 4: You divided the participants into two groups, NC=17 and CI=41, as we can see from Table 1. However, on the line 213 in Table 4 these numbers have been changed to NC=41 and CI=17. Which variant is correct?
- Table 1: Is the line named “Mann-Whitney U test” in Table 1 correct? As we can see from the lines 123-124 “…MoCA scores showed statistically significant difference among both groups (U=0.000, p<0.001)”, but in the line named “Mann-Whitney U test” in Table 1 we may see U=367.500 (p = 0.752) for MoCA scores.
- Table 1: The line with Standard Deviation for Mean Values should be added to Table 1.
- Lines 133-138: These lines should be removed because this information is presented in Table 2.
- Table 2: The line with Standard Deviation for Mean Values should be added to Table 2.
- Table 4: It is unclear whether you have revealed an increase or a decrease in the curvature in the CI group. Table 4 shows an increase in mean values in the CI group, but the Discussion tells us that there were reductions in mean cortical curvature in the group with cognitive impairment. You should clarify whether the higher mean values in Table 4 indicate a reduced cortical curvature or an increased one.
- In Conclusion you suggest that cortical curvature metrics could serve as early, sensitive biomarkers of cognitive decline. However, as we can see from the plots showing differences between groups (Figures 2 and 6), this method is only appropriate for population studies. Therefore, this metric cannot be used as biomarkers for individuals. Please comment on this.
- References: Multiple references to the same article (7,19, 23, 25, 40) were given. Is this correct?
Author Response
Dear Reviewer,
We appreciate the time and effort you have taken to review our manuscript. Your insightful comments and suggestions have been crucial in improving the clarity, accuracy, and overall quality of our work. Below, we provide a detailed response to each of your comments, along with the corresponding revisions made to the manuscript.
Line 27: It is unclear which group (CI or NC) had a lower mean curvature.
- The sentence has been paraphrased for clarity and specificity. The revised version now explicitly states the affected hemisphere and lists the regions with their corresponding statistical values:
"In the right hemisphere, statistically significant changes in mean curvature were observed in the isthmus cingulate (U = 188.5, p = 0.006), lingual (U = 202.5, p = 0.013), pars orbitalis (U = 221.5, p = 0.031), and posterior cingulate regions (U = 224.5, p = 0.035)."
Line 67: There is no reference to this author and article in the References.
- The missing reference has been corrected. The citation now correctly refers to de Moraes et al. in the references section.
Lines 87-89: It’s unclear how the understanding of morphometric alterations could potentially improve the intervention pathway. You should explain this phrase, or provide a
- Thank you, we have revised this section to provide a clearer explanation:
“A better understanding of these morphometric changes may help in complementing diagnostic criteria, enabling earlier detection of neurodegenerative disorders. As a result, this could facilitate more timely and targeted interventions, such as cognitive therapies or pharmacological treatments, as a result improving patient outcomes.”
Line 106: The aim should be specified. It should not be phrased as “further examine…”. It should be explained what were you trying to find exactly?
- The aim statement has been reworded to be more precise:
“In this context, this study aims to investigate whether mean cortical curvature can potentially serve as a biomarker for distinguishing cognitively impaired individuals from healthy controls. Specifically, we sought to determine if alterations in cortical curvature are associated with different levels of cognitive performance and whether these changes correlate with established measures of cognitive decline.”
Line 115: You divided the participants into two groups, but then you divided them into three groups. Could you explain why you did that?
- This has been clarified in the Methods section:
“Initially, participants were divided into two groups—cognitively impaired (CI) (MoCA score ≤ 25) and who did not have cognitive impairment – normal cognition (NC) (MoCA score ≥ 26)— to broadly assess differences in cortical curvature between individuals with and without cognitive impairment. However, given that cognitive decline occurs on a spectrum, a second classification was applied, dividing participants into three groups: normal cognition (NC), moderate cognitive performance (MPG), and low cognitive performance (LPG) based on MoCA scores. This finer categorization allowed for a more detailed analysis of cortical curvature changes across varying degrees of cognitive impairment, providing insights into potential alterations in brain morphology.”
Line 116: What about patients with 25 scores?
- This has been clarified, ensuring that the classification is inclusive of patients with a score of 25.
Lines 122-123: Does this phrase mean that there was no statistically significant difference in age?
- The text has been revised for clarity:
“To analyze differences in age and MoCA scores between the two groups, the Mann-Whitney U test was performed. The results indicated that there was no statistically significant difference in age between the groups (U = 367.500, p = 0.752). However, a statistically significant difference was found in MoCA scores (U = 0.000, p < 0.001), confirming distinct cognitive performance levels between the groups.”
Table 1, Table 4: You divided the participants into two groups, NC=17 and CI=41, as we can see from Table 1. However, on the line 213 in Table 4 these numbers have been changed to NC=41 and CI=17. Which variant is correct?
- Thank you, this inconsistency has been corrected to ensure that the participant numbers are accurately reported across all tables.
Table 1: Is the line named “Mann-Whitney U test” in Table 1 correct? As we can see from the lines 123-124 “…MoCA scores showed statistically significant difference among both groups (U=0.000, p<0.001)”, but in the line named “Mann-Whitney U test” in Table 1 we may see U=367.500 (p = 0.752) for MoCA scores.
- The correct values have been added to ensure consistency between the text and Table 1.
Table 1: The line with Standard Deviation for Mean Values should be added to Table 1.
- Standard deviation values have now been included in Table 1.
Lines 133-138: These lines should be removed because this information is presented in Table 2.
- These lines have been removed to avoid redundancy.
Table 2: The line with Standard Deviation for Mean Values should be added to Table 2.
- Standard deviation values have now been included in Table 2.
Table 4: It is unclear whether you have revealed an increase or a decrease in the curvature in the CI group. Table 4 shows an increase in mean values in the CI group, but the Discussion tells us that there were reductions in mean cortical curvature in the group with cognitive impairment. You should clarify whether the higher mean values in Table 4 indicate a reduced cortical curvature or an increased one.
- Upon review, we identified an error in the data presented in Table 4, where the values were incorrectly inserted. This issue has now been corrected to ensure accuracy. The revised table now correctly reflects the cortical curvature values for the NC and CI groups, aligning with the findings described in the Results and Discussion sections. We have also verified the consistency of these corrections throughout the manuscript to avoid further discrepancies.
In Conclusion you suggest that cortical curvature metrics could serve as early, sensitive biomarkers of cognitive decline. However, as we can see from the plots showing differences between groups (Figures 2 and 6), this method is only appropriate for population studies. Therefore, this metric cannot be used as biomarkers for individuals. Please comment on this.
- The conclusion has been modified to clarify that cortical curvature is more suitable for population-level studies rather than individual diagnosis:
“These findings reinforce the importance of cortical folding patterns in neurodegenerative disease progression. However, while cortical curvature metrics show significant differences at the group level, their variability limits their use as standalone biomarkers for individual diagnosis. Future research should investigate whether combining cortical curvature with other neuroimaging or cognitive biomarkers can enhance early detection at the individual level”
References: Multiple references to the same article (7,19, 23, 25, 40) were given. Is this correct?
- This has been reviewed and corrected to ensure proper referencing.
Thank you for your time and valuable feedback.
Sincerely,
Renāte Rūta Apse
Reviewer 3 Report
Comments and Suggestions for Authors
The manuscript titled “Morphometric Measurement of Mean Cortical Curvature: Analysis of Alterations in Cognitive Impairment” addresses an interesting subject. However, there are several areas where the authors can enhance the quality of the manuscript:
- Keywords: The keyword section is overly lengthy. It should be summarized and arranged alphabetically.
- Materials and Methods Section: Please include a detailed explanation of how the sample size for this study was determined.
- Results Section: The legends for Figures 3, 5, and 7 need further elaboration.
- Grammatical Errors: The manuscript contains many grammatical errors that need to be corrected to improve readability.
- Abbreviations: A comprehensive list of abbreviations used in the manuscript should be provided.
The manuscript contains many grammatical errors that need to be corrected to improve readability.
Author Response
Dear Reviewer,
We sincerely appreciate your feedback on our manuscript. Below, we provide detailed responses to your comments and outline the revisions made accordingly.
Keywords: The keyword section is overly lengthy. It should be summarized and arranged alphabetically.
The keyword section has been revised to be more concise and has been arranged in alphabetical order as per the suggestion.
Materials and Methods Section: Please include a detailed explanation of how the sample size for this study was determined.
The sample size for this study was determined based on the availability of patients recruited from ambulatory consultations. Participants were selected consecutively based on their eligibility criteria to ensure a representative sample of individuals with varying degrees of cognitive performance. While no formal power analysis was conducted prior to recruitment, the final sample size of 58 participants was deemed sufficient for statistical comparisons and aligned with similar neuroimaging studies investigating cortical curvature in cognitive impairment.
Results Section: The legends for Figures 3, 5, and 7 need further elaboration.
The figure legends have been expanded to provide more context about the anatomical regions. These revisions enhance clarity and ensure the results are understood by the reader.
Grammatical Errors: The manuscript contains many grammatical errors that need to be corrected to improve readability.
A grammatical revision has been performed throughout the manuscript. Issues related to sentence structure, clarity, and readability have been addressed to improve the overall quality of the text.
Abbreviations: A comprehensive list of abbreviations used in the manuscript should be provided.
A comprehensive list of abbreviations has been provided at the end of the manuscript. We have reviewed and ensured that all relevant abbreviations are included for clarity.
Thank you for your time and valuable feedback.
Sincerely,
Renāte Rūta Apse